# Paclitaxel Induces Upregulation of Transient Receptor Potential Vanilloid 1 Expression in the Rat Spinal Cord

**DOI:** 10.3390/ijms21124341

**Published:** 2020-06-18

**Authors:** Yukako Kamata, Toshie Kambe, Terumasa Chiba, Ken Yamamoto, Kazuyoshi Kawakami, Kenji Abe, Kyoji Taguchi

**Affiliations:** 1Department of Medicinal Pharmacology, Showa Pharmaceutical University, Machida, Tokyo 194-8543, Japan; distance.of.2254@gmail.com; 2Department of Pharmacology, Showa Pharmaceutical University, Machida, Tokyo 194-8543, Japan; kanbe@ac.shoyaku.ac.jp; 3Faculty of Pharmaceutical Sciences, Nihon Pharmaceutical University, 10281 Komuro, Ina-machi, Kitaadachi-gun, Saitama 362-0806, Japan; t-chiba@nichiyaku.ac.jp (T.C.); k-abe@nichiyaku.ac.jp (K.A.); 4Department of Education and Research Center for Clinical Pharmacy, Showa Pharmaceutical University, Machida, Tokyo 194-8543, Japan; k-yamamoto@ac.shoyaku.ac.jp; 5Department of Pharmacy, Cancer Institute Hospital, 3-10-6 Ariake, Koto-Ku, Tokyo 135-8550, Japan; kazu.kawakami@jfcr.or.jp

**Keywords:** paclitaxel, transient receptor potential vanilloid 1, spinal cord, peripheral neuropathic pain, TRPV1 antagonist, small interfering RNA

## Abstract

Painful peripheral neuropathy is a common adverse effect of paclitaxel (PTX) treatment. To analyze the contribution of transient receptor potential vanilloid 1 (TRPV1) in the development of PTX-induced mechanical allodynia/hyperalgesia and thermal hyperalgesia, TRPV1 expression in the rat spinal cord was analyzed after intraperitoneal administration of 2 and 4 mg/kg PTX. PTX treatment increased the expression of TRPV1 protein in the spinal cord. Immunohistochemistry showed that PTX (4 mg/kg) treatment increased TRPV1 protein expression in the superficial layers of the spinal dorsal horn 14 days after treatment. Behavioral assessment using the paw withdrawal response showed that PTX-induced mechanical allodynia/hyperalgesia and thermal hyperalgesia after 14 days was significantly inhibited by oral or intrathecal administration of the TRPV1 antagonist AMG9810. We found that intrathecal administration of small interfering RNA (siRNA) to knock down TRPV1 protein expression in the spinal cord significantly decreased PTX-induced mechanical allodynia/hyperalgesia and thermal hyperalgesia. Together, these results demonstrate that TRPV1 receptor expression in spinal cord contributes, at least in part, to the development of PTX-induced painful peripheral neuropathy. TRPV1 receptor antagonists may be useful in the prevention and treatment of PTX-induced peripheral neuropathic pain.

## 1. Introduction

Chemotherapy-induced peripheral neuropathy occurs with several chemotherapeutic agents, such as cisplatin, oxaliplatin, vincristine, and paclitaxel. Paclitaxel (PTX) is widely used for the treatment of solid tumors, including those of breast, ovary, and lung [1], and it has side effects that are a major factor limiting the quality of life of chemotherapy patients [2]. PTX-induced acute neuropathy usually presents as painful sensory neuropathy, with numbness, tingling, mechanical allodynia, and burning sensations on the feet and hands or both [3,4,5,6]. Recent evidence suggests that transient receptor potential vanilloid 1 plays a crucial role in PTX-induced painful neuropathy. Rodent models of PTX-induced peripheral neuropathy were developed to elucidate the pain mechanisms caused by PTX treatment [7,8,9,10]. Neurophysiological and biochemical studies show that the activation and sensitization of nociceptors plays a key role in neuropathic pain behavior following PTX treatment [11,12,13].

Transient receptor potential vanilloid 1 (TRPV1) is a non-selective cation channel with high permeability for Ca^2+^, which is primarily expressed in the spinal cord and peripheral terminals of non-myelinated primary afferent neurons [14,15]. TRPV1 is connected to a complex network of pain-related mechanisms [16]. TRPV1 receptors in sensory neurons are important for acute thermal nociception and inflammatory hyperalgesia [17,18,19], and they are also important in cancer chemotherapeutic drug-induced neuropathic pain [12,20]. The expression of TRPV1 was observed to increase in the spinal cord during inflammation [21,22]. Recently, many studies indicated that TRPV1 receptors in the spinal cord are involved in PTX-induced neuropathic pain [23,24]. Gene deficiency and pharmacological studies provided convincing evidence that TRPV1 significantly contributes to both neuropathic and chronic inflammatory pain [24,25]. TRPV1 was also shown to be involved in the pathogenesis of chemotherapy-induced peripheral neuropathy. Recent studies suggested an active role for TRPV1 receptors in the pathological mechanisms of PTX-induced peripheral neuropathy both in the dorsal root ganglion (DRG) and in the spinal cord [23,26,27]. In addition, TRPV1 receptors are considered an important factor in the mechanism underlying spinal cord synaptic modulation of nociceptive signaling during PTX-induced peripheral neuropathic pain. However, few studies considered the role of spinal TRPV1 upregulation in PTX-induced peripheral neuropathic pain.

This study investigated the involvement of TRPV1 in the spinal cord in PTX-induced painful peripheral neuropathy. In addition, we evaluated whether PTX-induced neuropathic pain was associated with increased TRPV1 expression in the spinal cord and if administration of a TRPV1 receptor antagonist alters PTX-induced peripheral neuropathic pain.

## 2. Results

### 2.1. Effect of PTX on Mechanical Allodynia/Hyperalgesia

As expected, the mean paw withdrawal frequency measured using the von Frey filaments (vFF) test was significantly increased seven and 14 days after the start of PTX treatment (2 and 4 mg/kg) compared to the vehicle (Figure 1A,B). On day 14, the responses to 2 g vFF stimulation with PTX were significantly increased (*p* < 0.01) by 64.0% ± 9.6% (2 mg/kg) and 79.9% ± 8.8% (4 mg/kg) compared to the vehicle (21.5% ± 2.1%, Figure 1A). Similar results were obtained in response to 5 g vFF stimulation (Figure 1B). On day 14, the responses to 5 g vFF stimulation with PTX treatment were further increased compared to vehicle treatment by 76.2% ± 9.6% (2 mg/kg) and 88.5% ± 8.9% (4 mg/kg) (*p* < 0.01) compared to vehicle treatment (24.2% ± 4.1%, Figure 1B). The results of the present study are consistent with our previously published findings [26]. Thus, PTX-treated rats developed mechanical allodynia/hyperalgesia.

### 2.2. Effect of PTX on Thermal Hyperalgesia

Mean paw withdrawal latency (s) was decreased, as determined by thermal stimulation, seven and 14 days after the start of PTX treatment (2 mg/kg or 4 mg/kg i.p.) compared to vehicle treatment (Figure 1C). PTX produced a significant decrease in paw withdrawal incidence (%) on day 14 compared to vehicle treatment; similarly, the responses to thermal stimulation were significantly decreased by 7.5 ± 0.7 s (2 mg/kg; *p* < 0.01) compared to vehicle treatment (9.7 ± 1.0 s). Similar results were obtained in response (6.5 ± 1.0 s; *p* < 0.01) to 4 mg/kg PTX (Figure 1C). Thus, this indicates that PTX treatment induces thermal hyperalgesia in rats.

### 2.3. Effect of PTX Treatment on TRPV1 Protein Expression in Spinal Cord

We removed the spinal cord at seven and 14 days after the start of PTX treatment (4 mg/kg). TRPV1 protein expression was quantified using Western blot analysis and compared with β-actin protein expression. As shown in Figure 2A, PTX (4 mg/kg) treatment significantly increased TRPV1 protein expression in the spinal cord at days 7 (138.2% ± 12.2%, *p* < 0.01) and 14 (153.5% ± 4.9%, *p* < 0.01). Thus, PTX significantly increased TRPV1 protein expression in the spinal cord. Immunohistochemistry revealed that the majority of TRPV1 protein expression in the spinal cord was localized to the superficial layers of the spinal dorsal horn (Figure 2B). Using computerized optical density (OD) image analysis, we found that PTX (4 mg/kg) induced a significant increase in TRPV1 protein expression in the superficial layers of the spinal dorsal horn at day 14 (130.0% ± 4.7%, *p* < 0.01) compared to the vehicle. Thus, PTX (4 mg/kg) increased TRPV1 protein expression in the superficial layers of the spinal dorsal horn.

### 2.4. Effect of Oral Administration of TRPV1 Antagonist (AMG9810) on PTX-Induced Mechanical Allodynia/Hyperalgesia

The effects of the TRPV1 antagonist AMG9810 on PTX-induced mechanical allodynia/hyperalgesia were investigated. Administration of AMG9810 (30 mg/kg, p.o.) significantly inhibited mechanical allodynia/hyperalgesia on day 14 after the PTX treatment (Figure 3). There was a significantly lower value (44.8% ± 3.5%, *p* < 0.05) at 60 min in the responses to 2 g vFF stimulation before AMG9810 administration (58.5% ± 8.9%, Figure 3A). More potent inhibition was obtained in response to 5 g vFF stimulation at 60 min (withdrawal response rate before AMG9810: 70.6% ± 9.3%, 60 min after AMG9810: 47.4% ± 6.3%, *p* < 0.01, Figure 3B). The effect of AMG9810 diminished by 180 min and withdrawal frequency reverted to control levels thereafter. However, there were no significant changes in withdrawal latency observed after AMG9810 administration in the Cremophor^®^ vehicle-treated rats (Figure 3A,B). Thus, these results suggest that blockade of TRPV1 receptors reduced PTX-induced mechanical allodynia/hyperalgesia.

### 2.5. Effect of Intrathecal Administration of TRPV1 Antagonist (AMG9810) on PTX-Induced Mechanical Allodynia/Hyperalgesia and Thermal Hyperalgesia

The role of spinal TRPV1 antagonists was investigated by examining the effects of the TRPV1 antagonist AMG9810 on PTX (4 mg/kg i.p.)-induced mechanical allodynia/hyperalgesia and thermal hyperalgesia (Figure 4A–C). Intrathecal administration of AMG9810 (35 μg/20 μL) significantly decreased mechanical allodynia/hyperalgesia and thermal hyperalgesia on day 14 after PTX treatment compared to before AMG9810 administration. The inhibiting effect of AMG9810 observed using the vFF test on mechanical allodynia (2 g) was weak compared to mechanical hyperalgesia (5 g, Figure 4A,B). In addition, PTX-induced thermal hyperalgesia, as evaluated by the paw withdrawal latency time, was significantly extended in the AMG9810 group at 60 and 90 min from post-administration (Figure 4C). The withdrawal latency returned to baseline levels within 180 min after intrathecal AMG9810 administration. However, there were no significant changes in withdrawal latency observed after intrathecal AMG9810 administration in Cremophor^®^ vehicle-treated rats (Figure 4A–C). Thus, the TRPV1 antagonist AMG9810 reversed PTX-induced mechanical allodynia/hyperalgesia and thermal hyperalgesia. These data indicate an important role of spinal TRPV1 receptor activation in PTX-induced mechanical allodynia/hyperalgesia and thermal hyperalgesia.

### 2.6. Effect of Intrathecal Administration of TRPV1 Small Interfering RNA (siRNA) on PTX-Induced Peripheral Neuropathy

Our results suggest that PTX-induced mechanical allodynia/hyperalgesia and thermal hyperalgesia are critically dependent on functional TRPV1 in the spinal cord. We, therefore, proposed that siRNA targeting TRPV1 would prevent PTX-induced peripheral neuropathy. Firstly, we confirmed that the TRPV1 protein levels in the spinal cord of the TRPV1 siRNA group were significantly lower than in the siRNA-negative control group (Figure 5). Next, rats were intrathecally treated with either targeting TRPV1 siRNA or siRNA-negative control for three days (for 72 h until the behavioral testing at day 14) in PTX (4 mg/kg) treatment. PTX-induced mechanical allodynia/hyperalgesia and thermal hyperalgesia in the TRPV1 siRNA-negative control group were not affected on days 7 and 14 (Figure 6A–C). In contrast, the TRPV1 siRNA group showed a significant decrease in PTX-induced peripheral neuropathy at day 14 compared to the siRNA-negative control group (Figure 6A–C). Thus, intrathecal administration of TRPV1 siRNA significantly increased the paw withdrawal mechanical threshold and thermal latency in rats administered PTX.

## 3. Discussion

Paclitaxel (PTX)-induced peripheral neuropathic pain can be a significant problem for cancer patients. We demonstrated that mechanical allodynia/hyperalgesia and thermal hyperalgesia were observed on day 14 after the start of 2 or 4 mg/kg PTX treatment, which was administered on four alternate days from days 0 to 6. Behavioral experiments showed that PTX treatment induced mechanical allodynia/hyperalgesia and thermal hyperalgesia in rats, which is an agreement with previous reports [28,29,30].

Transient receptor potential vanilloid 1 (TRPV1) is a non-selective cation channel with high permeability for calcium ions, and it is primarily expressed in the central and peripheral terminals of non-myelinated primary afferent neurons [31]. TRPV1 on peripheral nociception neurons is thought to act as a transducer and molecular integrator of peripheral nociceptive stimuli [32]. TRPV1 receptors could play a pivotal role in the modulation of nociceptive signaling in inflammatory pain [33]. TRPV1 is known to play an important role in peripheral neuropathic pain induced by chemotherapeutic drugs [10,20,34]. Thus, TRPV1 receptors in the spinal cord might be activated by PTX treatment.

In the present study, we demonstrated that PTX-treated rats developed mechanical allodynia/hyperalgesia and thermal hyperalgesia, and they showed increased TRPV1 protein levels in the superficial dorsal horn of the spinal cord. In addition, immunohistochemical analysis showed that TRPV1 expression was increased in the superficial dorsal horn of the spinal cord following PTX treatment. TRPV1 in the superficial dorsal horn of the spinal cord is present on the central branches of DRG sensory neurons [35]. Several studies suggested that TRPV1 receptors are expressed in the superficial dorsal horn of spinal cord, as well as being expressed in rat astrocytes and microglia [18,27,36,37,38]. We speculate that the mechanism underlying PTX-induced neuropathic pain may be attributable to TRPV1 receptor expression in astrocytes and microglia in the spinal cord. In addition, our previous study suggested that PTX treatment upregulates TRPV1 in small- and medium-diameter DRG neurons [26], which is then transported along their central axons to the superficial dorsal horn in the spinal cord. TRPV1 in the spinal cord is present on central branches of small and medium cells in DRG neurons [35,39]. An immunohistochemical study showed that there are direct contacts between TRPV1 containing afferents with neurokinin 1 (NK-1) positive dorsal horn neurons in lamina I [40]. We previously reported that PTX treatment increases the release of substance P, but not CGRP, in the superficial layers of the spinal dorsal horn, which may contribute to PTX-induced painful peripheral neuropathy [28]. Moreover, PTX treatment induced significant upregulation of nuclear c-Fos expression in superficial spinal cord horn neurons that was diminished by TRPV1 antagonists (SB366791 and AMG9810) [24]. The present data suggest that upregulation of TRPV1 expression in the superficial dorsal horn of the spinal cord may contribute to PTX treatment-induced enhancement of mechanical allodynia/hyperalgesia and thermal hyperalgesia. Thus, spinal TRPV1 receptors appear to play an important role in PTX-induced peripheral neuropathy. In our previous report, systemic administration of PTX resulted in mechanical allodynia/hyperalgesia, while administration of the TRP antagonist ruthenium red or the TRPV1 antagonist capsazepine completely reversed PTX-induced thermal hyperalgesia [26]. In the present study, we observed that either p.o. or intrathecal administration of AMG9810 reversed the increase in mechanical allodynia/hyperalgesia and thermal hyperalgesia of PTX-treated rats to the level observed in Cremophor^®^ vehicle-treated rats. The antihyperalgesic effect of AMG9810 proved to be more effective following intrathecal administration than oral administration. This suggests that direct blockade TRPV1 receptor in the spinal cord is an important factor for the observed effects of AMG9810. Furthermore, our data showed that PTX treatment upregulated TRPV1 receptor expression in the spinal cord. In addition, TRPV1 is reported to be upregulated on nociceptive Aδ and C-fibers [41,42], indicating that blockade of TRPV1 produces a greater impact on pain transmission during neuropathic pain than under normal conditions. Thus, TRPV1 receptors appear to play an important role in spinal cord pain transmission. It was reported that AMG 9810 does not significantly alter motor function, as measured by open-field locomotor activity and motor coordination tests [43]. In our experiments, the antihyperalgesic effect of AMG9810 is not likely to be due to sedative effects, because no significant effects of AMG9810 were observed on rearing behavior, investigatory behavior, and sniffing (data not shown). Similarly, intrathecal application of AMG9810 was also shown to attenuate PTX-induced hypersensitivity [34]. Intrathecal administration of selective TRPV1 receptor antagonists reduced mechanical allodynia in a model of chronic constriction injury and inflammatory thermal hyperalgesia [44,45,46]. Moreover, AMG9810 was observed to reverse increased sensitivity to heat and mechanical stimuli in rats [47]. It is possible that TRPV1 antagonists inhibit PTX-induced mechanical allodynia/hyperalgesia and thermal hyperalgesia by blocking TRPV1 receptors. Therefore, our data support the hypothesis that PTX induces activation of TRPV1 receptors in the spinal cord, thereby inducing peripheral neuropathic pain.

In order to confirm the observed effects of PTX-induced peripheral neuropathic pain, we employed an siRNA targeting TRPV1. Intrathecal administration of TRPV1 siRNA decreased the induction of TRPV1 protein in the spinal cord and inhibited mechanical allodynia/hyperalgesia and thermal hyperalgesia induced by PTX treatment. In contrast, rats treated with the siRNA-negative control showed no alterations in mechanical and thermal hypersensitivity induced by PTX treatment. Whether TRPV1 siRNA can reach the spinal cord in sufficient concentrations following intrathecal delivery is frequently questioned. However, several reports demonstrated that intrathecally delivered TRPV1 siRNA accumulates in the spinal cord [48,49]. TRPV1 siRNA was administered continuously for three days using an osmotic pump, and we expect that TRPV1 siRNA levels in the spinal cord were sufficient to reduce TRPV1 receptor expression. Previously, we showed that intrathecal administration of TRPA1 antisense using an osmotic pump, but not TRPA1 mismatched oligodeoxynucleotides, knocked down TRPA1 expression in the DRG and decreased oxaliplatin-induced cold hyperalgesia [50]. Furthermore, immunochemical analysis revealed that siRNA treatment significantly reduced TRPV1 expression. The crucial involvement of TRPV1 in driving inflammatory pain is well established through the use of TRPV1 knock-out mice [51,52]. Moreover, a previous behavioral study showed that intrathecal administration of antisense TRPV1 reduced mechanical hypersensitivity in rats with spinal nerve ligation [53]. Thus, TRPV1 receptors play an important role in the in synaptic transmission of PTX-induced peripheral neuropathic pain information in the spinal cord.

## 4. Materials and Methods

### 4.1. Experimental Animals

Male Wistar rats weighing 250 to 320 g were used in the present study. All rats were housed individually under automatically controlled environmental conditions with a 12 h light/dark cycle (lights on from 8:00 a.m. to 8:00 p.m.) and provided free access to food and water. All animals were quarantined in a centralized animal facility for at least seven days upon arrival. Each animal was used only once. Experiments were carried out according to the guidelines for animal care and use published by the National Institutes of Health and the committee of Showa Pharmaceutical University (No: p-2013-1; date 22 March 2016).

### 4.2. Drug Administration

Rats were treated with PTX as described previously [12,22]. In brief, PTX (2 or 4 mg/kg per mL, prepared with saline from TAXOL^®^, Bristol-Myers-Squibb, 6 mg/mL PTX in EL vehicle) or Cremophor^®^ vehicle (Cremophor EL-polyethoxylated castor oil and ethanol, diluted with two parts saline to one part Cremophor EL; Sigma-Aldrich, St. Louis, MO, USA) was administered intraperitoneally (i.p.) on four alternate days (days 0, 2, 4, and 6 of the experiment; cumulative doses of 8 or 16 mg/kg). The volume injected depended on the animal’s weight to ensure i.p. or p.o. injections of less than 2.5 mL. Oral administration of AMG9810 (30 mg/kg, Tocris Bioscience, Ellisville, MO, USA) or intrathecal administration of AMG9810 (35 μg/20 μL) was conducted on day 14 of the experiment. AMG9810,

(*E*)-3-(4-*t*-butylphenyl)-*N*-(2,3-dihydrobenzo[b][1,4]dioxin-6-yl) acrylamide, is reported as a TRPV1 antagonist with high selectivity [43]. Rats were injected intrathecally with AMG9810 using a 25 µL Hamilton syringe with a 28-gauge needle. Rats showing motor impairment were excluded from further analysis.

### 4.3. Behavioral Observation (Mechanical or Thermal Stimulation)

Observers blinded to the experimental conditions performed mechanical behavioral testing on the same time zone (10:00 a.m. to 3:00 p.m.) on days 0, 7, and 14. In brief, rats were placed in a plastic box (20 × 18 × 13 cm) with a wire grid floor and allowed to habituate to the environment for 20 min. In mechanical stimulation, the sensitivity of the plantar surface of the hind paw was measured as the withdrawal responses to von Frey filaments (vFF). Filaments of two forces (2 and 5 g) were applied to the mid-plantar surface of both hind paws. In ascending order of force, each filament was applied to each hind paw five times, with each application held for 5 s. A 1-min rest was allowed between tests on alternate hind paws and 3–4 min between subsequent tests on the same hind paw. A positive response was recorded if the paw was withdrawn during the application of the vFF or immediately after its removal. Withdrawal responses to the vFF from both hind paws were counted and expressed as an overall percentage response, e.g., if a rat withdrew three out of the total 10 vFF applications, this was recorded as a 30% overall response to the vFF.

In thermal stimulation, we used a radiant thermal stimulator (IITC Life Science, Woodland Hills, CA, USA), and the intensity of the light was adjusted at the start of the experiment so that average baseline latencies were approximately 8–10 s with a cut-off latency of 20 s. The focus of radiant thermal stimulation was precisely positioned on the middle of the plantar surface of each hind paw, and the latency to paw withdrawal was recorded. Three trials were performed at intervals of 5 min, and one score was assigned for each session by averaging the last two trials. The latencies were obtained alternately from the left and right hind paws, which were tested 5 min apart. The data were reported as mean values for both the right and left hind paws combined. Radiant thermal stimulation was performed before PTX administration (day 0) and on days 7 and 14 after the first dose.

### 4.4. Western Blot Analysis

The rats were deeply anesthetized with pentobarbital (60 mg/kg, i.p.) on days 0, 7, or 14 after the start of PTX (4 mg/kg) treatment, and spinal cords were removed for Western blot analysis. Spinal cords were homogenized in cold 1× RIPA lysis buffer (Santa Cruz Biotechnology, Santa Cruz, CA, USA) containing phosphatase inhibitor cocktail (EDTA free) (Nacalai Tesque, Inc. Tokyo. Japan) and protease inhibitor cocktail (Nacalai Tesque). The homogenates were centrifuged at 4 °C for 30 min at 15,000× *g*, and the supernatants were collected. Spinal cord protein (30 μg) was separated by an SDS-polyacrylamide gel (7.5%) and transferred onto polyvinylidene difluoride membranes. Anti-TRPV1 antibody (1:200, Alomone Labs, Jerusalem, Israel) was used for Western blotting, with anti-β-actin (Sigma-Aldrich, 1:1000) used as the internal control. Horseradish peroxidase-labeled goat anti-rabbit immunoglobulin G (IgG) (Santa Cruz Biotechnology, 1:5000) was used as the secondary antibody. Specific bands were detected by enhanced chemiluminescence (ECL) with the ECL^TM^ Prime Western Blotting Detection kit (GE Healthcare, Wauwatosa, WI, USA) using a luminescent image analyzer LAS4000 (Fuji Film, Japan). The intensities of immunoreactive bands were quantified using MultiGage Ver.3 software (Fuji Film, Tokyo, Japan).

### 4.5. Immunohistochemistry

The rats were deeply anesthetized with pentobarbital (60 mg/kg, i.p.) on day 14 after the start of PTX (4 mg/kg) treatment, and spinal cords were prepared for immunohistochemistry. Rats were perfused transcardially with 20 mL of potassium-free phosphate-buffered saline (K^+^-free PBS; pH 7.4) followed by 50 mL 4% paraformaldehyde solution. The spinal cord (L_4–6_) was removed, post-fixed for 3 h, and cryoprotected overnight in 25% sucrose solution. The spinal cord was stored at −80 °C until use. The spinal cord sections were cut at 10 μm thickness, thaw-mounted on silane-coated glass slides, and air-dried overnight at room temperature. Spinal cord sections were incubated with excess blocking buffer containing 2% skim milk in 0.1% Triton X-100 in K^+^-free PBS, then subsequently reacted overnight at 4 °C with anti-TRPV1 antibody (Alomone Labs, 1:200) in 2% bovine serum albumin/0.1% Triton X-100 in K^+^-free PBS. The sections were then incubated in fluorescein isothiocyanate-conjugated anti-rabbit IgG (Sigma-Aldrich, 1:200) for 2 h at room temperature. All sections were treated with Permafluor (Thermo Shandon, Pittsburgh, PA, USA), cover-slipped, and evaluated with microscopy.

### 4.6. Image Analysis

The immunostained sections were observed with an Olympus laser-scanning confocal microscope (FLUOVIEW BW50, Olympus, Tokyo, Japan) at wavelengths of 488 and 568 nm. Signals were analyzed under fluorescence microscopy at 400× magnification using a digital camera system. Cells were counted in a blinded manner by experimenters unaware of the experimental protocol. A total of five sections (90 μm apart) were randomly selected from each spinal cord. The optical density (OD) of the superficial layers of the spinal cord was calculated for day 14 after PTX or vehicle treatment. OD of the stained sites was determined with the public domain ImageJ program.

### 4.7. Intrathecal Injection of TRPV1 siRNA or TRPV1 Antagonist (AMG9810)

Under sodium pentobarbital (50 mg/kg, i.p.) anesthesia, the rat atlanto-occipital membrane was cut. A soft tube (Silascon, Kaneka Medix Company, Osaka, Japan; outer diameter, 0.64 mm) was inserted into the subarachnoid space for a length of 8.0 cm to ensure that the tip reached the lumbar enlargement. The rostral part of the catheter was sutured to the occipital muscle to immobilize the catheter, and the wound was closed in two layers with 3-0 silk thread. For TRPV1 knockdown, the siRNAs specific for TRPV1 (5′–GCGCAUCUUCUACUUCAACTT–3′) and the siRNA universal negative control (5′–CAACUUCAUCUUCUACGCGTT–3′) were purchased from Sigma-Aldrich, and 0.5 nmol of each was prepared in PBS and mixed with artificial cerebrospinal fluid (aCSF). To obtain sustained infusion of siRNA targeting TRPV1 or siRNA-negative control (0.5 nmol/μL/h), an ALZET^®^ osmotic pump (seven-day pump, 1 μL/h; DURECT) was filled with TRPV1 siRNA or siRNA-negative control in aCSF. The TRPV1 siRNA (Sigma Genosys siRNA Service) was intrathecally administered on day 11 to day 14 (for 72 h until the behavioral testing at day 14) with PTX treatment. Whether TRPV1 siRNA can reach the spinal cord in sufficient concentrations following intrathecal delivery is frequently questioned. However, several reports demonstrated that intrathecally delivered siRNA accumulates in the spinal cord [48,49]. On the other hand, AMG9810 was administered orally (30 mg/kg, p.o.) or intrathecally (35 μg/20 μL of aCSF with 20% DMSO) before the mechanical or thermal stimulation test 14 days after PTX treatment. Intrathecal administration of AMG9810 was performed via the same catheter. Rats showing motor impairment were excluded from further analysis.

### 4.8. Statistical Analysis

All data are expressed as the mean ± SEM. For two-group comparisons, data were calculated using the *F*-test, followed by Student’s or Aspin Welch’s *t*-test. Multiple comparison test, e.g., vehicle versus PTX 2 mg/kg versus 4 mg/kg treatment group, was calculated using one-way analysis of variance (ANOVA) followed by Dunnett’s multiple comparisons test. In evaluation of the effect of AMG9810, the significance was measured between before and after AMG9810 treatment using the above-mentioned multiple comparison test. Statistical significance was accepted at *p* < 0.05.

## 5. Conclusions

Taken together, these findings suggest that the activation of TRPV1 receptors in the dorsal horn of the spinal cord following PTX treatment may be, at least in part, related to the activation of TRPV1 receptors and may be involved in the development of PTX-induced peripheral neuropathic pain.

## Figures and Tables

**Figure 1 ijms-21-04341-f001:**
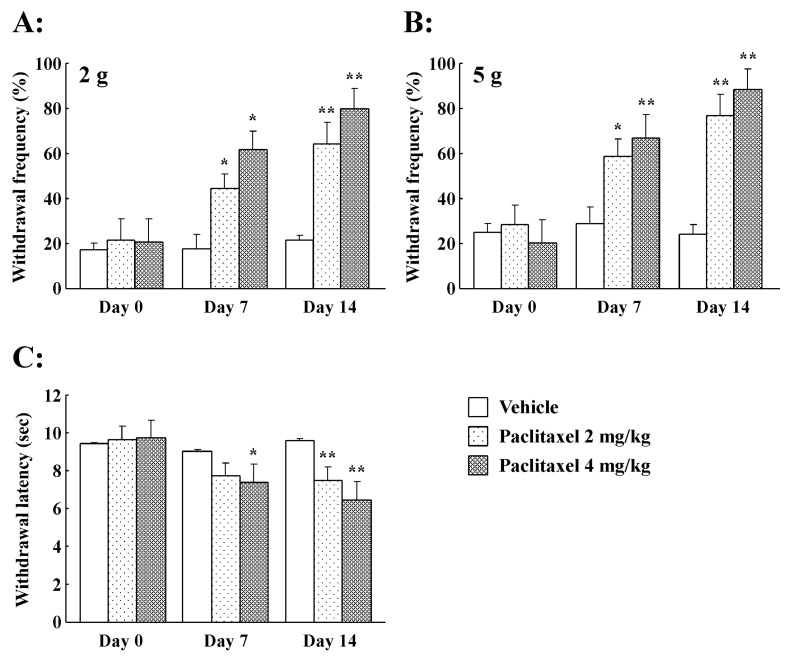
Paclitaxel (2 and 4 mg/kg, i.p.)-induced mechanical allodynia/hyperalgesia and thermal hyperalgesia as determined by the von Frey test and paw thermal stimulation test, respectively. (**A**) von Frey filaments (2 g) and (**B**) von Frey filaments (5 g) were used to measure mechanical allodynia/hyperalgesia induced by paclitaxel treatment (administered on days 0, 2, 4, and 6) in rats. Histograms show the withdrawal frequency on days 7 and 14 after the start of paclitaxel or Cremophor^®^ vehicle treatment. (**C**) The paw thermal stimulation test was used to measure thermal hyperalgesia induced by paclitaxel treatment in rats. Histograms show the withdrawal latency at days 7 and 14 after the start of paclitaxel and Cremophor^®^ vehicle treatment. Data are the mean ± standard error of the mean (SEM) of *n* = 5–6 rats. * *p* < 0.05, ** *p* < 0.01 from vehicle treatment group.

**Figure 2 ijms-21-04341-f002:**
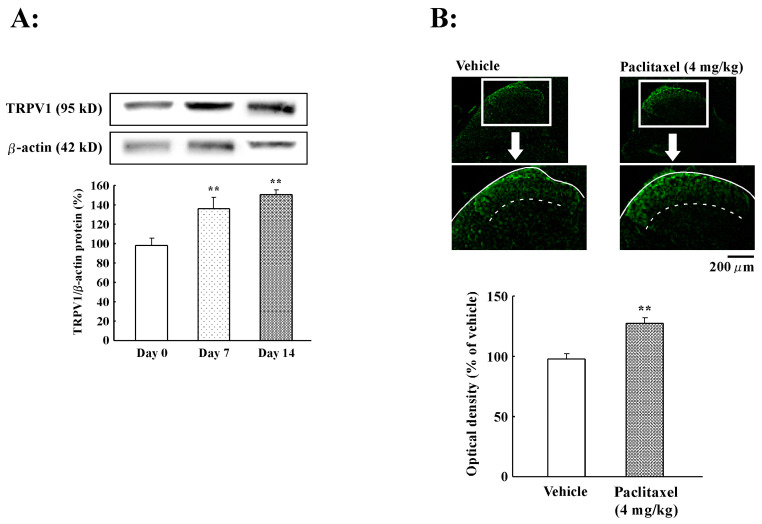
Expression of transient receptor potential vanilloid 1 (TRPV1) protein in rat spinal cord. (**A**) Western blotting analysis of the effect of paclitaxel on TRPV1 protein expression (representative blot shown). TRPV1 protein in the spinal cord at days 0, 7, and 14 after the start of paclitaxel treatment (4 mg/kg) was measured. TRPV1 protein expression was normalized to β-actin expression. Paclitaxel treatment increased TRPV1 protein expression. (**B**) Immunohistopathological analysis of the effect of paclitaxel on TRPV1 protein expression. Paclitaxel (4 mg/kg) induced a significant increase of TRPV1 protein expression. The histogram shows the relative amount of TRPV1 protein in paclitaxel (4 mg/kg)-treated rats. Data are the mean ± SEM of *n* = 4–5 for the paclitaxel group. ** *p* < 0.01 versus day 0 or vehicle treatment group.

**Figure 3 ijms-21-04341-f003:**
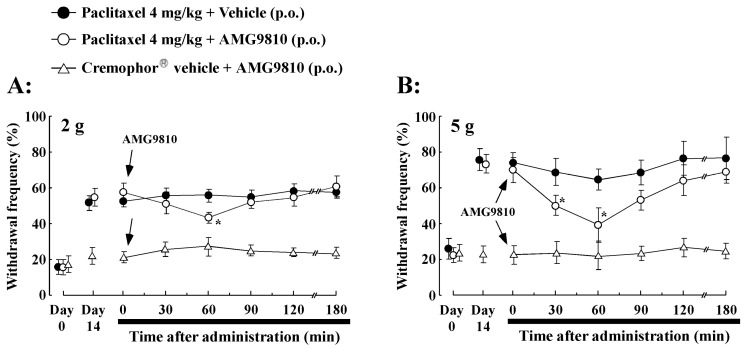
Effect of oral administration of a TRPV1 antagonist (AMG9810, 30 mg/kg) on paclitaxel-induced mechanical allodynia/hyperalgesia. (**A**) von Frey filaments (2 g) and (**B**) von Frey filaments (5 g) was used to measure mechanical allodynia/hyperalgesia (paw withdrawal frequency) induced by 4 mg/kg of paclitaxel treatment in rats. Oral administration of AMG9810 was conducted at day 14. Paclitaxel-induced mechanical allodynia/hyperalgesia was significantly inhibited by AMG9810. Data are the mean ± SEM of *n* = 8–10 rats. * *p* < 0.05 from before AMG9810 administration.

**Figure 4 ijms-21-04341-f004:**
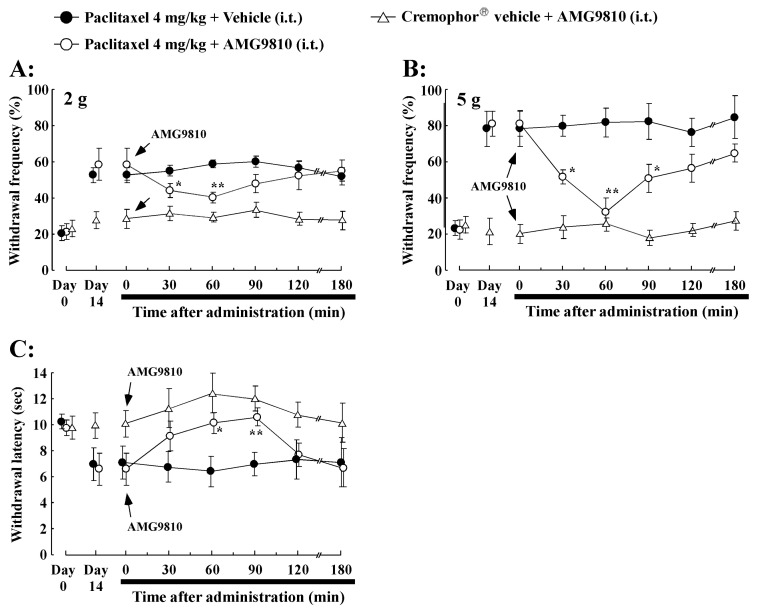
Effect of intrathecal administration of TRPV1 antagonist (AMG9810, 35 μg/20 μL) on paclitaxel-induced mechanical allodynia/hyperalgesia and thermal hyperalgesia. (**A**) von Frey test (2 g), (**B**) von Frey test (5 g), and (**C**) paw thermal test were used to measure mechanical allodynia/hyperalgesia and thermal hyperalgesia induced by 4 mg/kg paclitaxel treatment in rats, respectively. Intrathecal administration of AMG9810 (35 μg/20 μL) or Cremophor^®^ vehicle was conducted at day 14. Paclitaxel-induced mechanical allodynia/hyperalgesia and thermal hyperalgesia were significantly inhibited by AMG9810. Data are the mean ± SEM of *n* = 8–10 rats. * *p* < 0.05, ** *p* < 0.01 from before AMG9810 administration.

**Figure 5 ijms-21-04341-f005:**
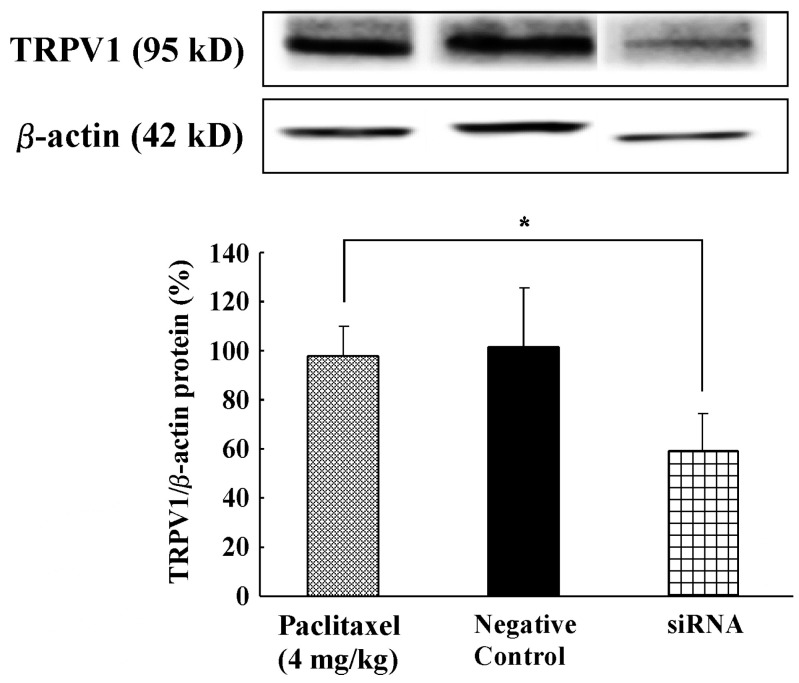
Confirmation of TRPV1 small interfering RNA (siRNA) effects on TRPV1 protein levels in the spinal cord. TRPV1 siRNA reduced TRPV1 protein expression. TRPV1 and β-actin protein in the spinal cord at day 14 after paclitaxel treatment (4 mg/kg) were measured. Treatment with TRPV1 siRNA significantly reduced TRPV1 protein levels compared to the paclitaxel treatment group. The top panel shows a representative Western blot in which TRPV1 and protein expression was normalized to β-actin expression as expressed relative to the paclitaxel (4 mg/kg) treatment group. The histogram shows the effect of paclitaxel treatment and TRPV1 siRNA treatment on relative amounts of TRPV1 protein expression. Data are the mean ± SEM of *n* = 5–6 rats. * *p* < 0.05 relative to the paclitaxel group.

**Figure 6 ijms-21-04341-f006:**
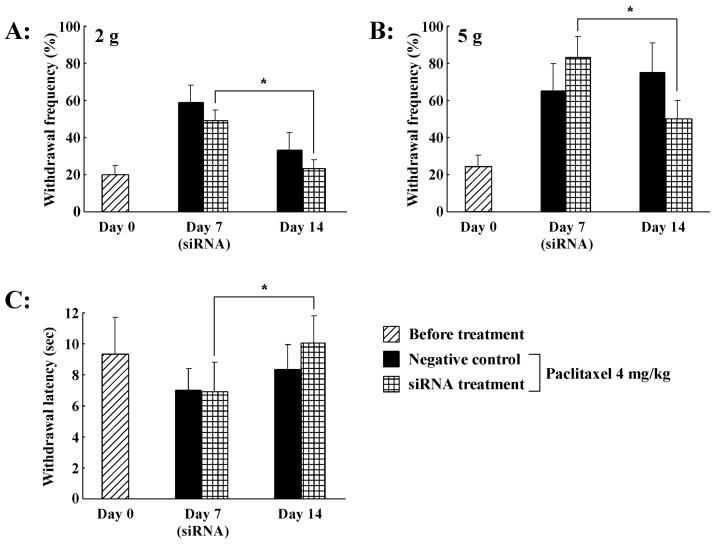
Effect of intrathecal administration of TRPV1 siRNA on paclitaxel-induced peripheral neuropathy. The TRPV1 siRNA or siRNA-negative control was intrathecally administered on days 11 to 14 after the start of paclitaxel treatment. (**A**) Paclitaxel-induced mechanical allodynia and (**B**) Paclitaxel-induced hyperalgesia, and (**C**) Paclitaxel-induced thermal hyperalgesia were significantly reduced in the TRPV1 siRNA group at day 14 relative to the siRNA-negative control group. Histograms show withdrawal responses at days 0, 7, and 14 after the start of paclitaxel treatment. Data are the mean ± SEM of *n* = 5–6 rats. * *p* < 0.05 in statistical comparison between days 7 and 14.

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
