# Peer review of "Paclitaxel Induces Upregulation of Transient Receptor Potential Vanilloid 1 Expression in the Rat Spinal Cord"

_ijms, 2020, doi:10.3390/ijms21124341_

Round 1

Reviewer 1 Report

  1. This resubmitted manuscript has many errors in figures and results. Figure legends did not match with figures.
  2. The originality of this manuscript is low. The authors should identify which cells mainly express TRPV1 in the spinal cord after paclitaxel treatment (e.g. primary afferent terminals, spinal cord neurons, glia...).

Reviewer 2 Report

There are no further comments to be adressed.

This manuscript is a resubmission of an earlier submission. The following is a list of the peer review reports and author responses from that submission.

Round 1

Reviewer 1 Report

The study of Kamata et al. investigates the role of spinal TRPV1 in a model of neuropathic pain induced by paclitaxel. Studies that have shown the involvement of the TRPV1 in paclitaxel-induced neuropathic pain and its up-regulation already exist and therefore the innovative character of the paper could be represented by the effect of spinal TRPV1 blocking / knocking down on mechanical allodynia/hyperalgesia and thermal hyperalgesia. My criticisms are listed below

Majors:

A reference indicating the selectivity of AMG9810 should be added. How the dose of AMG9810 was chosen? The description of the method for oral administration is missing. What was the effect of AMG9810 in sham mice/rats?

Another problem concerns the choice of b-actin as the internal control of the WB. There are changes in b -actin expression so it is difficult to evaluate changes in TRPV1 protein expression.

It is not clear if the authors are using rats or mice because both are mentioned in the text.

“Whether TRPV1 siRNA can reach the spinal cord in sufficient concentrations by intrathecal delivery has been debated”. It is not clear whether the authors refer to previous studies or the current one; however, the appropriate references or discussion of the case in the discussion are missing.

In the Discussion, there is no comment on the effect of AMG9810 which is greater when administered intrathecally. What is the contribution of spinal TRPV1 when the drug is administered systemically? Does AMG9810 pass the blood-brain barrier?

It is impossible to compare paclitaxel with oxaliplatin data on the TRPV1 expression because the former was measured after 7 and 14 days, while the latter after only 2 days.

Introduction

“The putative non-selective cation channel, transient receptor potential vanilloid 1 (TRPV1), which is activated by capsaicin”… Capsaicin is just one of the compounds/stimuli that activates the receptor.

 Lines 49-52: These sentences seem to contradict each other and must be corrected.

Results

Lines: 84-85: “Results for thermal hyperalgesia were similar to those in response to von Frey filament stimulation” This sentence is unclear and needs to be improved

Lines 101-102: “The peak response value (60 min after AMG9810 administration) of paclitaxel (4 mg/kg) was 44.8 ± 101 3.5% (2 g) and 40.5 ± 9.9% (5 g), which was significantly lower (P < 0.05) than vehicle treatment (57.0 102 ± 4.0% (2 g) and 66.9 ± 5.9% (5 g)).” It is not clear whether it refers to the paclitaxel or AMG9810 peak and needs to be clarified. Consider using the square bracket instead of the double round one.

Line 103: “Mice administered AMG9810 showed a dose-dependent decrease”: Mice that received the administration of….

Discussion

Lines 220-223: “In an electrophysiological study, administration of the TRPV1 antagonists capsazepine and AMG9810 reduced spontaneous excitatory postsynaptic current frequency recorded in lamina 2 neurons [36]. Therefore, we suggest that paclitaxel induces activation of TRPV1 receptors in the spinal cord, leading to peripheral neuropathic pain”. The conclusions that the authors propose do not seem appropriate

Reviewer 2 Report

The study by Kamata et al. nicely describes the role or spinal expressed TRPV1 in the processing of paclitaxel-induced neuropathy. The authors show, that TRPV1 expression is upregulated in the spinal cord of rats after paclitaxel treatment and furthermore that inhibition of TRPV1 by AMG9810 or knockdown of TRPV1 by siRNA attenuates paclitaxel-induced hyperalgesia/allodynia. The study is well designed, however I have some comments which should be addressed by the authors:

Major:

  • What is the effect of AMG9810 in the dose used in motor coordination and balance. These data should be presented or at least discussed.
  • Expression of TRPV1 after treatment with oxaliplatin in the spinal cord was investigated 2 days after treatment in contrast to paclitaxel (here, TRPV1 expression was investigated 7 and 14 days after treatment and fluorescence staining were performed only 14 d after paclitaxel treatment. This time point difference seems to be quite large. Please provide expression data for TRPV1 at later time points after oxaliplatin treatment  and furthermore also quantify expression levels via Western Blot, as immunofluorescence signals may already differ by slightly different isolation and incubation steps.
  • I would suggest presenting data in a different order, in order to allow the reader to follow the study a little better. I would present the expression data of TRPV1 after paclitaxel treatment (Fig. 4) after Fig. 1 (introduction/ presentation of the model) and before presenting data using the TRPV1 antagonist. Moreover I would also switch the order of Fig. 5 and 6 – meaning: I would present expression data of TRPV1 after siRNA knockdown first to proof that this experimental approach was successful and afterwards the behavior data.

Minor:

  • Line 172: delete extra “.” after “on days 7 and 14”

Reviewer 3 Report

This manuscript shows that TRPV1 receptors in the spinal cord may have a role in mediating paclitaxel-induced neuropathic pain in rats. However, more experimental works should be done before publication. I only suggest several points as follows;

1. Originality/Novelty is low. In addition, the rational for studying spinal TRPV1 is not presented in the Introduction. Most descriptions seem to be about peripheral TRPV1 in the Introduction. 

2. It is unclear which cells' TRPV1 receptors play a key role in paclitaxel pain. DRG cells? Spinal lamina I neurons? Spinal astrocytes? Substantia Gelatinosa neurons? Figure 4 images seem to suggest the possibility of astrocytes.

3. Confocal images in Figure 4 have low quality. Confocal microscope should provide higher resolution images. I recommend the authors to consult with a technical staff of Microscopy room. In addition, please try double or triple immunofluorescence staining to define the major cell types of TRPV1 up-regulation.

4. In Figure 4C, IHC was done at 2 days after an oxaliplatin injection. Where are the behavioral data (mechanical/thermal hypersensitivity) for these rats? Oxaliplatin also induces peripheral neuropathic pain in rats. 

5. Minor points: The line 137 and 143 should be different. "last dose" in line 303 may be "first dose".